# The Role of the Immune System in Cutaneous Squamous Cell Carcinoma

**DOI:** 10.3390/ijms20082009

**Published:** 2019-04-24

**Authors:** Matthew J. Bottomley, Jason Thomson, Catherine Harwood, Irene Leigh

**Affiliations:** 1Transplantation Research and Immunology Group, Nuffield Department of Surgical Sciences, University of Oxford, Oxford OX3 9DU, UK; 2Centre for Cell Biology and Cutaneous Research, Barts and the London School of Medicine and Dentistry, Queen Mary University of London, London E1 2AT, UK; jason.thomson@doctors.net.uk (J.T.); caharwood@doctors.org.uk (C.H.); i.m.leigh@qmul.ac.uk (I.L.)

**Keywords:** malignancy, cancer, immunology, leucocyte, skin, cutaneous, immunity, SCC, cSCC, squamous cell carcinoma

## Abstract

Cutaneous squamous cell carcinoma (cSCC) is the second most common skin cancer. In immunosuppressed populations it is a source of considerable morbidity and mortality due to its enhanced recurrence and metastatic potential. In common with many malignancies, leucocyte populations are both protective against cancer development and also play a role in ‘sculpting’ the nascent tumor, leading to loss of immunogenicity and tumor progression. UV radiation and chronic viral carriage may represent unique risk factors for cSCC development, and the immune system plays a key role in modulating the response to both. In this review, we discuss the lessons learned from animal and ex vivo human studies of the role of individual leucocyte subpopulations in the development of cutaneous SCC. We then discuss the insights into cSCC immunity gleaned from studies in humans, particularly in populations receiving pharmacological immunosuppression such as transplant recipients. Similar insights in other malignancies have led to exciting and novel immune therapies, which are beginning to emerge into the cSCC clinical arena.

## 1. Introduction

Cutaneous squamous cell carcinoma (cSCC) represents the second most common skin cancer in the United States, with an increasing incidence of over 700,000 cases each year [1]. In the great majority of cases, excision with clear margins is curative. Classical risk factors for cSCC development include age, ultraviolet (UV) radiation exposure, ethnicity, skin phototype, and immunocompromise.

Understanding of the role of immunity in tumor development generally has evolved over time and has been reviewed in depth elsewhere [2]. The concept that premalignant and oncogenic virally infected cells bearing immunogenic tumor-specific antigens are detected and eliminated by leucocytes through immune surveillance was superseded by the more complex idea of immunoediting. Immunoediting describes the process by which elimination of tumor cells leads to positive selection for those that have lost expression of immunogenic tumor-specific antigens, which proliferate and develop a tumor microenvironment (TME). Immunoediting has been further refined into a three-step process: elimination, equivalent to the original concept of ‘immune surveillance’; equilibrium, in which immune responses control but fail to fully eliminate the sporadic malignant cells that escape destruction, leading to ‘sculpting’ of the remaining cells; and escape, in which the selected cells are able to grow in an immunologically intact host, through a combination of immune evasion and immune suppression or resistance to apoptosis. 

There is no reason why this model should not apply to cSCC. However, the picture emerging is complex: leucocyte sub-populations may play contrasting roles at different stages of cSCC carcinogenesis. Infiltrating leucocytes of both innate and adaptive origin may play a role in tumor antigen clearance, but certain subpopulations may be subverted and reprogrammed in vivo to then suppress an immune response or directly promote tumor proliferation. Further complicating matters, inflammatory mediators often act in an autocrine and paracrine fashion upon keratinocytes, malignantly transformed cells and leucocytes, with differing effects upon each.

In this review, we discuss the current understanding of the role of individual elements of the innate and adaptive immune system in the development of, and response to, cSCC. We then discuss lessons learned about the role of the immune system in the clinical setting, with a particular focus on studies in the immunosuppressed. Finally, we explore emerging data about how the immune system itself may be harnessed to provide novel therapies against cSCC.

## 2. The Role of Innate Immunity

### 2.1. Tumor-Associated Macrophages (TAMs) and Neutrophils

Cross-talk between injured keratinocytes and resident macrophages may represent the triggering event for circulating leucocyte recruitment leading to the development of chronic inflammation, which plays a key role in cSCC development [3].

UVB-associated inflammation is marked by myeloid infiltration, angiogenesis, and keratinocyte hyperproliferation in a toll-like receptor 4 (TLR4)-dependent manner in mice [4]. Proinflammatory mediators are released by keratinocytes and infiltrating leucocytes in response to UVB irradiation or endotoxin stimulation [3,5,6,7]. These mediators, particularly tumor necrosis factor α (TNFα), play a critical role in early cSCC development in murine models [5,8,9]. Low levels of TNFα in the TME may lead to monocyte differentiation into a macrophage phenotype that promotes local tumor growth, angiogenesis, and immune escape [10].

In the setting of established malignancy, macrophages are one of the major tumor-infiltrating leucocyte populations. In vitro and in animal models, primed macrophages have been demonstrated to eradicate various (non-cSCC) tumor cell lines [11,12,13,14]. Despite this, across multiple malignancies the presence of gene signatures associated with intratumoral myeloid populations are associated with a poorer prognosis [15]. This paradox may be explained by TME reprogramming of myeloid cells. Tumor-associated macrophages (TAMs) infiltrating later in cSCC carcinogenesis are likely influenced by paracrine signaling, including TGF-β and relatively low concentrations of IFN-γ and TNFα, promoting a humoral Type 2 T helper (Th2)-M2 (as opposed to cytotoxic Type 1 T helper (Th1)-M1) response leading to ineffective phagocytosis and antigen presentation with acquisition of an immunoregulatory phenotype [16,17,18]. 

Macrophages have been identified around and within excised cSCC in greater numbers than normal skin with evidence of both M1 and M2 macrophage phenotypes [19,20]. TAMs may directly promote tumor growth through secretion of proangiogenic factors such as vascular endothelial growth factor (VEGF) and matrix metalloproteases (MMP). These degrade the peritumoral scaffold that might otherwise act as a barrier to invasion [19,21]. 

### 2.2. Dendritic Cells

Dendritic cells patrol peripheral tissue and migrate to regional lymph nodes upon stimulation, where antigen is efficiently presented particularly through Major Histocompatibility Complex (MHC) classes I and II, initiating an adaptive response in antigen-specific T cells through cognate binding and appropriate co-stimulatory signals. This determines the initial nature and strength of an immune response, driving an inflammatory response that facilitates clearance of tumor, or a tolerogenic response that may promote tumor growth through induction of unresponsiveness (‘anergy’) or apoptosis in T cells [22]. Most understanding of the role of dendritic cells (DCs) in cSCC comes from murine models. The best-described subtypes of skin-resident DCs are Langerhans cells (LCs) and monocyte-derived dermal DCs (dDCs), though up to four other DC-progenitor-derived subtypes within the dermis have been described [23,24]. LCs are the only DC found in the epidermis during steady state and are long-lived and self-renewing [25]. 

UV radiation induces a functional and quantitative reduction in systemic immune responses. UV radiation induces both DC apoptosis and lymph node migration, leading to depletion in the skin [26,27,28,29]. However, both lymph node migration and antigen presentation are impaired by UV radiation leading to a reduction in DC-mediated inflammation and promotion of Th2 and regulatory T cell responses [29,30,31,32,33]. dDCs may be differentially activated compared to LCs, more efficiently promoting cytotoxic responses partly through cross-presentation to CD8^+^ T cells [34]. 

Large-scale, genome-wide, and transcriptome-wide association studies have identified certain MHC Class II (HLA-DR and -DQ) single nucleotide polymorphisms (SNPs) that may be associated with increased cSCC risk in humans [35]. The effects of these have not been elucidated mechanistically but could relate to altered antigen presentation by DC to effector cells.

Whilst reduced numbers of LCs have been described in excised cSCC and Bowen disease (SCC in situ) compared to normal skin, it is difficult to interpret the significance of this, given that many of the effector functions of these cells lie elsewhere and may indicate enhanced migration to draining lymph nodes [20,36]. 

Constitutive absence of LC in chemically-induced cSCC appears to inhibit inflammation and mutagenesis required for tumor initiation and progression [37,38]. Conditional depletion of LCs and dDCs has conversely suggested an anticarcinogenic role though TNFα-mediated natural killer (NK) cell recruitment and activation [39,40]. DCs dampen inflammation initially in UVB-damaged murine skin through phagocytosis of apoptotic keratinocytes [41], but they subsequently promote tumor progression in an IL-22- and TGFβ1-dependent manner [31,42]. Despite mouse data to suggest the opposite, human LCs from established cSCC induce a strong T helper 1 (Th1) response in vitro even in the presence of tumor supernatant, whilst dermal DCs and monocyte-derived DCs show impaired function in the same setting [43,44].

Taken together, environmentally-induced altered DC function could facilitate early immune evasion and promote tumor development. LC dysfunction may not be sustained once the TME is established, unlike dDC dysfunction, which may be more important in the failure to mount adequate cytotoxic responses.

### 2.3. Myeloid-Derived Suppressor Cells (MDSCs)

The tumor milieu is able to pathologically induce immature myeloid precursor differentiation into a heterogeneous population of both monocytic and granulocytic lineages that demonstrate immunosuppressive properties and leads to reduced immunosurveillance and immune responses, termed myeloid-derived suppressor cells (MDSCs) [45]. MDSC accumulation within multiple malignancies has been associated with a poorer prognosis [46].

Little is known about the role of MDSC in cSCC. The generation and suppressive activity of MDSC in chemically-induced cSCC may be IL-6 dependent [47]. Various strains of human papillomavirus (HPV), which may be implicated in cutaneous cSCC development, have been shown to induce MDSC [48]. Based on other malignancies it is presumed that MDSCs inhibit antitumor responses and promote cSCC development.

### 2.4. Natural Killer (NK) and Innate Lymphoid Cells

Innate lymphoid cells (ILCs) belong to the lymphoid lineage but lack T- or B- cell receptors and do not express myeloid or dendritic cell markers. They are subdivided with ILC1 associated with Th1 responses, ILC2 with Th2, and ILC3 with Th17 responses [49].

NK cells are the best understood ‘cytolytic’ ILC1 population. Multiple localized subsets can be identified using cell surface markers, which recognize and kill tumor cells through killer receptors [50,51]. Patrolling NK cells are found in the human and murine dermis but rarely in the epidermis [52]. 

NK cells play a critical role in suppressing cSCC development in murine models [39], though the TME may modulate their cytotoxicity through downregulation of activating receptors [53]. Genetic or acquired impaired function or reduced number of NK cells is associated with increased risk of herpesvirus, papillomavirus, and cSCC in humans [54,55,56,57]. This indicates that NK cells play a key role in immunosurveillance against cSCC development.

Some NK cell populations may demonstrate features of immunological memory, including persistently enhanced function following activation. This appears to be particularly important in response to chronic, latent viral infection, and in vivo these populations may show enhanced antitumor activity [58,59]. It is possible, though unexplored, that chronic viral infections such as HPV may induce alterations in this process. 

Other noncytotoxic ‘helper’ ILC1, ILC2, and ILC3 populations have been described, but their role in cancer is more controversial [60]. ILC2 and ILC3 populations may promote tumor growth and spread through cytokine effects (particularly through IL-33 and IL-22), but they may also promote the formation of tertiary lymphoid structures that promote cytolytic adaptive responses [49,61]. In health, ILCs are preferentially found in ‘barrier’ tissues such as the skin (in contrast to NK cells, which are predominantly blood-borne) and have been demonstrated to play a role in cutaneous inflammation. Their role in cSCC carcinogenesis has yet to be explored [62].

### 2.5. Atypical T Cells

Two populations of T cells (γδ T cells and natural killer T cells) that display attributes of both innate and adaptive immunity are described. They express somatically rearranged receptors but may lack potential for establishing antigen-specific clonal memory populations. Both are able to secrete effector cytokines and engage in cell-mediated killing, akin to NK and CD8^+^ T cells. 

Gamma-delta (γδ) T cells are a population of predominantly CD3^+^CD4^−^CD8^−^ and CD3^+^CD4^−^CD8^+^ T cells bearing an atypical T cell receptor (TCR) containing a gamma and a delta chain that recognizes cells that present unprocessed, nonpeptide phosphorylated compounds [63]. These compounds are generated naturally under conditions of stress and are also found within microbial structures. In a recent analysis of the gene expression profile of thousands of samples from thirty-nine distinct human cancers, intratumoral γδ cell signatures correlated with a favorable prognoses [15]. Subsets of human γδ T cells have been defined based on use of one of two variable regions of TCR-δ [64]. Mice have different subsets of γδ T cells, of which the dendritic epidermal γδ T cells (DETCs) are the best studied [64]. 

DETCs patrol the skin, though they are still significantly outnumbered by their αβ T cell counterparts [40]. γδ cell knockout has demonstrated their role in protection against chemical-induced cSCC carcinogenesis in both a killer cell receptor- (NKG2D) and IgE-dependent manner [40,65]. More recent work suggests a potential tumor-promoting role for γδ cells in a murine model of HPV-induced epithelial malignancy, where IL-17A-producing γδ subsets were associated with increased angiogenesis [66]. γδ cells appear to represent only a minor subset of tumor-infiltrating lymphocytes, and their role within human cSCC is unknown. 

Natural killer T (NKT) cells represent 2%–3% of splenic T cells in mice and express intermediate levels of the classical αβ TCR. They express a restricted TCR repertoire and recognize glycolipid and nonlipid antigens, which are expressed on the nonclassical MHC Class I-like antigen-presenting molecule CD1. These cells efficiently suppress and activate both innate and adaptive immune responses [67].

NKT cells have been indirectly associated with a pro-carcinogenic role in UVB-induced cSCC. CD1d knockout leads to reduced UVB-induced inflammation, carcinogenesis, and a lack of functional NKT cells [68,69]. Adoptive transfer of splenic T cells from chronically irradiated mice leads to a failure to reject transplanted cutaneous malignancy—an effect lost when NKT cells were depleted [70]. Like γδ cells, their role in cSCC development in humans remains unclear.

## 3. The Role of Adaptive Immunity

### 3.1. Cellular Immunity

T cells play a major role in the immune response to tumor antigens. Up to 10% of tumor-infiltrating cells are T cells, both within the tumor and at tumor margins [71].

Activation of T cells is driven and directed by antigen presenting cells, leading to polarization of T cell responses. Multiple ‘axes’ of CD4^+^ T cell activation and differentiation have been described, including Th1, Th2, Th9, and Th17 (summarized in Table 1). A recent review excellently described the contrasting roles different Th subclasses generally play in the development of malignancy [71].

Specifically regarding cSCC, tumor-specific T cell responses are critical for prevention of cutaneous malignancy development and metastasis; these are generally cytotoxic responses mediated via Th1 pathways [76,77]. Irradiation and chemical cSCC models induce Th17 responses, which promote cSCC development in murine knockout models [78,79,80]. UVB irradiation inhibits effector CD4^+^ and CD8^+^ T cell responses against DNA damage in murine models [74]. 

IL-22 is a hallmark of Th22 responses, and it is also seen in benign skin conditions characterized by increased keratinocyte turnover. This cytokine is upregulated in cSCC and drives tumor proliferation in vitro [81]. T cell ‘crawl out’ analysis from explanted cSCC from organ transplant recipients (OTRs) found increased Th22 expression compared to cSCC from nonimmunosuppressed individuals [81].

T follicular helper (Tfh) cells provide support within germinal centers for the development and maturation of antigen-specific B cell responses and antibody production [75]. Emerging data suggests that Tfh-dendritic cell interactions may play a critical role in generating a germinal center reaction to intradermal antigens [82]. Whilst these cells are presumed to be integral to the development of B cell responses in cSCC (see ‘humoral immunity’, below) their role specifically has not been explored. 

Counterbalancing ‘effector’ T cell responses are ‘regulatory’ T cell (Treg) populations. Whilst multiple regulatory T cell populations have been identified, the canonical Treg population in humans possesses the phenotype CD4^+^CD25^hi^CD127^lo^FOXP3^+^ [83,84,85,86]. 

Treg are found within the tumor bed of most malignancies, including cSCC, and their frequency is often correlated with a poorer prognosis [87,88,89,90]. Upon activation, Treg inhibit effector responses through a number of direct and indirect mechanisms. They act upon effector cells, preventing cytokine secretion and proliferation, and antigen presenting cells, where they reduce the quality of antigen presentation or alter the balance of costimulatory signaling towards inhibition of effector activation. This is summarized in the setting of malignancy in Figure 1.

UV irradiation leads to cutaneous infiltration of Treg and induction of Treg from infiltrating T cells. Later in cSCC carcinogenesis, TGF-β is produced by malignant cells, which can induce infiltrating effector T cell conversion to Treg in other malignancies [91,92]. Treg are found in greater numbers in cSCC compared to normal skin, and they are more frequent in moderately and poorly differentiated cSCC compared to well-differentiated lesions [91]. Knockout of IL-10 in murine models prevents UV-induced carcinogenesis by impairing Treg function [76]. 

### 3.2. CD8^+^ T Cells

CD8^+^ T cells are essential for elimination of tumors in animal models [93,94,95], classically in parallel with Th1 responses, suggesting a key role in immunosurveillance. CD8^+^ T cell knockout leads to increased cSCC burden in chemical and irradiation models [78,80]. Surprisingly, a transgenic model using an HPV oncogene failed to find a protective role for CD8^+^ T cells against cSCC development [96].

Within the established TME, infiltrating CD8^+^ T cells often fail to clear tumors due to a number of mechanisms that lead to anergy and exhaustion (Figure 1). Within metastatic cSCC, inhibitory receptors (and potential markers for exhaustion) including Tim-3, cytotoxic T-lymphocyte–associated antigen 4 (CTLA-4), and programmed death-1 (PD-1) are frequently found on CD8^+^ T cells [97]—the presence of their ligands within cSCC, specifically PD-L1, is associated with poorer outcomes [98]. Local signaling may induce CD8^+^ T cells to develop a pathogenic role within cSCC, promoting tumor proliferation [81,99]. Infiltrating CD8^+^ T cells can be induced to demonstrate regulatory functions within other malignancies [100,101]. 

### 3.3. Humoral Immunity

B cells undertake effector functions through immunoglobulin (antibody) production as well as antigen presentation and production of effector cytokines, which contributes to polarization of T cell responses [102,103,104]. It was, therefore, long-held that B cells played an antitumor effector role. However, populations of B cells with regulatory properties ex vivo have been described in both animals and humans recently [105,106,107,108,109]. Thus, B cells, in common with other leucocyte populations, may play a role both in cancer progression and cancer immunity [110].

In animal models of chemically and genetically driven cSCC, knockout of B cells leads to a failure of carcinogenesis. This may be due to the proinflammatory and chemotactic role played by B cell-produced immunoglobulin, with stromal immune complex deposition early in carcinogenesis leading to myeloid activation via Fc receptors [111,112]. B cells may also directly suppress the immune response to cSCC in a TNFα- and IL-10-dependent manner [113]. B cell-derived IgE, produced in response to chemical carcinogenesis, may protect against cSCC development by promoting γδ responses in mice [114]. This suggests that the role of B cells may depend on the specific mutagen used to trigger carcinogenesis and the resultant dominant antibody class response.

In humans, an increased proportion, but not absolute number, of memory cells within the B cell compartment is independently associated with an increased risk of subsequent cSCC in OTRs [115]. Within established tumors, there is a reduced density of CD20^+^ B cells in OTRs compared to cSCC from the general population. How either of these findings translate mechanistically is unclear, particularly as many B cell immune functions are mediated at a distance from the tumor, such as in regional lymph nodes and in the bone marrow.

## 4. Lessons Learned from The Clinical Setting

### 4.1. Lessons Learned from Immunosuppressed Individuals

Systemic immunosuppression arising iatrogenically or through systemic disease is a significant risk factor for cSCC development. In the context of immunosuppression, cSCC are frequently multiple and tend to be more aggressive with a much higher burden of disease [116,117,118,119,120,121,122]. This provides compelling evidence for a role of the immune system in cSCC, and studying immunosuppressed populations has provided important insights into how the immune system contributes to cSCC development and progression. These immune mechanisms are complex, involving the interplay of reduced immunosurveillance/immunoediting, immunosenescence, the dynamic anti-tumor and pro-tumor influences of specific immune cells within the TME, and possibly a role for oncogenic viruses.

With an overall risk for cSCC up to 200-fold greater than age-matched, immunocompetent populations, OTRs are the best studied of the immunosuppressed populations with regards to cSCC [116]. cSCC is the most common post-transplant malignancy, and it exhibits a more aggressive disease course with a metastatic risk double that of the general population [123]. Median survival after metastasis is only two years [117,118,119,120,121,123].

cSCC burden in OTRs strongly correlates with increasing time after transplantation and intensity of immunosuppression, providing evidence for a dose-dependent effect on cSCC pathogenesis and prognosis [122,124,125]. Correlating with the intensity of immunosuppression needed to prevent allograft rejection, the highest risk is seen in cardiac and lung transplant recipients, then pancreas and/or kidney recipients, then recipients of liver transplants [125]. Increasing age at transplantation is also a strong predictor for the development of further cSCC and may correlate with increased duration of immunosuppression but also age-related changes in immunity (see below) [126,127]. Reducing the intensity of immunosuppression may reduce the risk of cSCC development, providing the rationale for modest dose reduction in patients who develop numerous cSCCs [124,128]. 

Populations immunosuppressed due to systemic disease also exhibit moderately increased cSCC risk, providing more evidence that it is immunosuppression driving this phenomenon, rather than other potential confounders such as host cancer risk factors and lifestyle factors. This includes chronic lymphocytic leukemia (8× increased risk) and non-Hodgkin lymphoma (5× increased risk) [117,129,130,131,132]. Patients with HIV infection have a 2.6-fold increase in cSCC risk, which correlates inversely with CD4 count [129,133]. Patients with rheumatoid arthritis immunosuppressed with methotrexate for disease control have a two- to four-fold increased risk of cSCC development compared to the general population [134], and exposure to thiopurines in patients with inflammatory bowel disease is associated in most studies with a significantly increased cSCC risk of up to five-fold [135]. The significantly elevated cSCC risk in OTRs specifically may be explained by the greater intensity of continuous immunosuppression with multiple agents, as well as the direct tumorigenic effects of certain agents (discussed below) [136].

### 4.2. The Effects of Immunosuppression on Immune Phenotype and Cutaneous Squamous Cell Carcinoma (cSCC) Risk

As discussed earlier, the role of immunity in cSCC development centers on the concepts of immunosurveillance and immunoediting. In the context of immunosuppression, dysfunction and suppression of specific leucocyte subsets peripherally and at the level of the TME impacts on these overall immune functions, resulting in the observed increased cSCC risk. 

Alterations in various circulating leucocyte populations have been indirectly linked to the increased aggressiveness of cSCC in immunosuppressed individuals [55,115,137,138,139]. This work has particularly focused on OTRs. Early studies suggested that a low peripheral blood CD4^+^ count predicted OTRs at increased risk of cSCC; however, the patients included were within their first 10 years post-transplantation where cSCC incidence is relatively low and has not been consistently replicated [55,140,141]. More recent studies identified that greater numbers of peripheral blood CD4^+^CD25^hi^CD127^lo^FOXP3^+^ Tregs are found in OTRs with a history of cSCC, and may also predict OTRs at increased risk of cSCC recurrence [55,115]. However, this predictive value may be relatively short-term and only for high-risk cSCCs [55,115]. The proportion of circulating T cells with a fully demethylated ‘Treg specific demethylated region’ within the *FOXP3* promoter (TSDR, proposed as a more specific marker of ‘true’ Treg) was greater in OTRs with a history of cSCC [137]. Functional studies have demonstrated that preservation of a peripheral blood Th1 effector response against tumor antigens (quantified by IFN-γ production) may be associated with reduced susceptibility to cSCC in OTRs [138].

OTRs with previous cSCC have also been observed to have lower overall numbers of B cells, with class-switching from naïve to memory phenotype observed [115]. Low numbers of NK cells are also associated with an increased cSCC risk in OTRs, although these observations are likely to be most relevant in patients on azathioprine, which is known to reduce numbers of both NK and B cells [139].

CD57 has been identified as an accurate marker of T cell senescence, expressed on terminally differentiated effector T cells that may display impaired proliferation and reduced effector cytokine production [139]. Stratification by CD57 expression on circulating CD8^+^ T cells identified OTRs at almost three-fold increased risk of developing subsequent cSCC after correction for potential confounders, a marker superior to most clinical indicators [139]. It is postulated that excess immunosuppression may promote T cell senescence through recurrent episodes of subclinical latent viral reactivation (e.g., cytomegalovirus, human papillomavirus, and Epstein–Barr virus) and subsequent inflammation, which over time leads to repeated rounds of antigenic stimulation and the accumulation of oligoclonally expanded senescent T cells. However, this has not been demonstrated directly [139]. Additionally, accumulation of CD57^+^ cells also correlates with loss of CD4^+^ and CD8^+^ central memory T cells, another important source of antitumor immunity [94]. Overall, immunosuppression may result in a reduced T cell antigen repertoire and impaired immunosurveillance, which promotes cSCC development and progression through immune evasion, one of the key hallmarks of cancer [142].

### 4.3. The Effects of Immunosuppression on the Tumor Microenvironment

Interactions between malignant and nonmalignant host cells constitute the TME, which is driven by complex, dynamic intercellular communications via networks of chemokines, cytokines, growth factors, and inflammatory and matrix remodeling enzymes [143]. Several nonmalignant cell types are found in the TME, including leucocytes, cells of the vasculature and lymphatics, fibroblasts and other cells of the stroma. The roles of these cells, their regulation, and their effects on tumor progression have been reviewed extensively elsewhere [143,144,145]. Cellular and molecular phenotyping of the TME in various cancers, in particular the immune infiltrate, have provided important insights into antitumor immune responses and tumor escape. This has improved our understanding of the role of the immune system in carcinogenesis, particularly in the context of immunosuppression [144]. Immunophenotyping has led to the identification of specific subclasses of immune TME that have varying effects on tumor initiation and can be used as biomarkers to predict response to immunotherapy [146].

In established cSCC, quantifying infiltrating leucocytes has consistently demonstrated a reduced density of intra- and peritumoral immune cell infiltrates in the context of chronic immunosuppression compared to nonimmunosuppressed controls, specifically CD4^+^ and cytotoxic CD8^+^ T cells [55,119,147,148]. In contrast, and reflecting what is observed peripherally, Treg numbers appear to be increased in the TME in immunosuppression [55,81,138]. The frequency of FOXP3^+^ Tregs in cSCC correlates with primary tumors that metastasize and overall poorer clinical outcomes [149]. Antigen presentation capacity in the TME is reduced in immunosuppression-related cSCC with reduced numbers of CD123^+^ plasmacytoid dendritic cells (pDCs) observed across the spectrum of cSCC neoplastic progression, with consequent reduction in signaling of IFN-γ, the prototypical Th1 cytokine [148]. In immunocompetent humans there is a mix of Th1- and Th2-associated gene expressions within established cSCC. However, within the cSCC of immunosuppressed individuals there is a skew away from Th1 towards Th2-associated gene expression and cell infiltration. There is also evidence of a reduction in expression of some but not all Th17-associated genes, though there does not appear to be a reduction in intratumoral Th17 cells, compared to non-immunosuppressed controls [81,150]. There is now mounting evidence that a predominantly T helper 2 (Th2) polarized microenvironment results in cSCC progression, and this is the microenvironment that appears to develop in the context of chronic pharmacological immunosuppression [78,150].

Our understanding of how immunosuppression affects the leucocytes in the TME is complicated by the heterogeneity in treatment regimens used in OTRs. Moreover, data on the pro- and anti-carcinogenic effects of the different immunosuppressant drugs is often conflicting and difficult to discern from epidemiological studies [116]. Each has different mechanisms of action and are given in combination to minimize the risk of allograft rejection [151]. Some agents have direct mutagenic effects that contribute to increased cancer risk. For example, azathioprine causes UVA photosensitivity, increased UVB-induced mutagenesis and recent genomic analysis has identified a novel azathioprine-associated mutational signature in cSCC that reveals a likely interplay with transcription-coupled nucleotide excision repair [152,153]. The main classes of immunosuppressant medications used in OTRs, their mechanisms of action, and effects on the immune system and tumor biology are summarized in Table 2 below.

### 4.4. A Role for Viruses in Cooperation with the Immune System?

An infectious contributor to cSCC development has long been hypothesized given its similar increased incidence compared to other immunosuppression-associated cancers caused by oncogenic viruses such as Epstein–Barr virus (EBV, post-transplant lymphoproliferative disorder) and human herpes virus 8 (HHV-8, Kaposi sarcoma) [116]. Indeed, genetic defects that result in susceptibility to persistent cutaneous infection by human β-papilloma viruses are associated with greatly increased risk of cSCC development [163]. Mechanistic studies in keratinocytes have demonstrated the ability of viral oncogenes to cause failure of DNA repair and apoptosis in cooperation with UV radiation, cell-cycle dysregulation and transformation whilst epidemiological studies have consistently demonstrated an association between β-PV and keratinocyte cancer, particularly in immunosuppression [164,165,166]. However, transcriptome analysis has failed to demonstrate that the HPV is actively replicating within cSCC. To date, there is insufficient evidence to confirm an unequivocal causal role, and this remains an area of controversy [167]. Overall, HPV burden may be more critical than specific HPV type [168], and impaired viral effector responses may indirectly play a role in carcinogenesis. 

### 4.5. Therapeutic Modulation of the Immune Response

Uncoupling the mechanisms of T cell co-stimulation has provided important insights into how tumors evade the immune system at the level of the TME, heralding a revolution in immunotherapy that has transformed cancer care and outcomes for advanced and metastatic disease [169]. Studying the TME across many cancers has identified upregulation of immune checkpoint co-signaling proteins such as cytotoxic T-lymphocyte-associated antigen 4 (CTLA-4) and programmed death-1 (PD-1), which effectively act as brakes on antitumor immune responses [169,170]. Ipilimumab, an anti-CTLA-4 monoclonal antibody, was the first-in-class checkpoint inhibitor that demonstrated a clear survival advantage in metastatic melanoma and was FDA-approved in 2011 [171]. Since then, antibodies inhibiting the PD-1/PD-L1 immune checkpoint, such as nivolumab and pembrolizumab, have demonstrated a more favorable safety profile and improved efficacy [172].

Tumors with a high mutation burden respond best to checkpoint inhibitor immunotherapy, which is in part thought to be due to their increased neoantigen load and corresponding immunogenicity [173,174]. As a primarily UV radiation-driven tumor, cSCC is one of the most highly mutated cancers with a mutational load of around 50 mutations per megabase of DNA, which is higher than melanoma and second only to basal cell carcinoma [175]. There have been anecdotal case reports of anti PD-1 treatment in advanced cSCC, and a recent phase 2 study of cemiplimab (an anti-PD-1 monoclonal antibody) in patients with locally advanced and metastatic cSCC demonstrated a response rate of 47% and durable disease control in 61% [176]. Clinical trials with other anti-PD-1 agents such as pembrolizumab are in progress (clinicaltrials.gov)

In a recent study using immunohistochemistry to profile PD-1/PD-L1 expression in the cSCC TME, PD-1 expression was found on 80% of CD8^+^ T cells, 73% of CD4^+^ T cells, and PD-L1 expression was found on 26% of tumor cells [177]. Immunosuppression did not appear to alter this phenomenon. Overexpression of PD-L1 in cSCC in the absence of a significant CD8^+^ T cell infiltrate was associated with increased metastatic risk and was also identified in lymphatic metastatic disease providing further evidence for the positive prognostic role of peritumoral CD8^+^ T cells [178]. In OTRs, the CTLA-4 and PD-1 axes have an important role in influencing the balance between Treg induction and effector T cell function to restrain or promote allograft responses [170]. There have been published case reports of OTRs suffering acute graft rejection and subsequent graft failure following treatment with anti-PD-1 therapy, including one patient with advanced cSCC [179,180,181]. Consequently, organ transplantation is a relative contraindication for checkpoint inhibitor therapy, and ORTs are excluded from clinical trials at present. Ultimately, the transplant patient with advanced cSCC may be left with the unpalatable choice of a shorter life expectancy with a functioning transplant or a longer life expectancy but with the potential of losing their organ transplant.

Adoptive cell transfer using chimeric antigen receptor T cells (CAR-T cells) is a novel, highly personalized immunotherapy that has shown impressive efficacy in hematological cancers, and it is currently licensed for certain childhood leukemias [182]. Extracted and expanded T cell populations are bio-engineered to exhibit specificity for tumor surface antigens before re-infusion back into the patient for tumor elimination [183]. CAR-T therapy for solid tumors is currently undergoing clinical trials and may represent a potential future treatment option for cSCC with less risk of triggering ‘off target’ immune responses to a transplant [182].

## 5. Conclusions

The immune system plays a key role in cSCC pathogenesis and progression that is multifaceted and complex. Transgenic animal models have provided important insights into the roles of specific leucocytes in keratinocyte cancer biology, but they are limited by differences in murine-human immune and tumor phenotypes and simplistic cancer induction models. Studying cSCC in immunocompetent and immunosuppressed individuals has uncoupled many of the immune mechanisms at play in cSCC pathogenesis and progression. In humans there is unlikely to be a single ‘trigger’ for cSCC development per se, with a combination of genetic and environmental (including viral, chemical, and radiation) factors culminating in malignant transformation in keratinocytes. It is not clear how current models can ‘tease out’ which are the dominant effects, and thereby the most likely role of individual leucocyte lineages, in cSCC. 

The clinical visibility of cutaneous premalignant lesions not only permits early diagnosis and excision where required but also provides an opportunity to understand the early events that precede the development of overt cSCC. In this regard, interrogation of both premalignant and malignant tissue in greater depth may facilitate greater understanding of the proteomic, genomic, and immunological landscape.

Intensive study of the TME across cancers including cSCC has begun uncoupling the complex, dynamic interactions at play between tumors and the multitude of host innate and adaptive immune cells and the mechanisms of immune evasion that evolving cancers employ. This improved knowledge is driving forward a revolution in immunotherapies that has changed the landscape for metastatic cancer therapy, providing novel treatment options for advanced and metastatic cSCC. However, our current understanding of the role of immunity in cSCC is far from complete, with conflicting data from many animal and human studies. Part of the problem arises from limitations in our ability to accurately profile leucocytes at high resolution, with their different activation states, spatial orientations, and multitudes of receptors, compounded by significant tumor and immune heterogeneity. Novel techniques are now becoming mainstream such as mass imaging cytometry and multiplex gene expression analysis, which promise accurate, high-throughput, and deep immune phenotyping with spatial resolution that will hopefully overcome these limitations and herald a new era in cancer immunology. As always with such technologies that produce high dimensional data, interpreting and interrogating the output will likely be the rate-limiting step. The need for computational tools and algorithms to decipher the complex interactions of the TME to bring about translational benefit to patients will be essential.

## Figures and Tables

**Figure 1 ijms-20-02009-f001:**
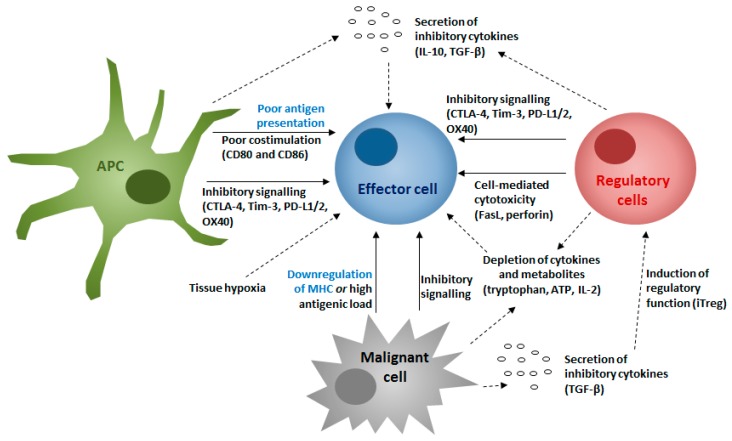
Mechanisms contributing to regulation of tumor clearance by effector cells. APC, antigen presenting cell. In this context, an APC may be a macrophage, dendritic cell (or Langerhans cell) or B cell. Regulatory cells may represent various populations, including Treg, B cell regulatory populations, and myeloid-derived suppressor cells (MDSCs). Effector cell relates predominantly to CD8^+^ T cells, but may also include natural killer (NK), natural killer T (NKT), gamma-delta T, and cytotoxic CD4^+^ T cells. The above mechanisms contribute to effector, particularly CD8^+^, cell dysfunction and lead to a failure of tumor clearance. Those mechanisms highlighted in blue represent mechanisms of immune escape for the tumor. Examples of relevant mediators for each pathway are provided in brackets. Solid arrows represent mechanisms directed against specific cells through cell contact, whilst dashed arrows represent nonspecific mechanisms through remodeling of the tumor microenvironment (TME).

**Table 1 ijms-20-02009-t001:** Summary of described T cell polarization axes and their role in cutaneous squamous cell carcinoma (cSCC). In italics is a summary of the role of each axis in cSCC development. IL, interleukin; IFN, interferon; TGF, transforming growth factor; TNF, tumor necrosis factor; and CCL, chemokine ligand.

Axis	Cytokines Driving Differentiation	Hallmark Effector Cytokines	Postulated Evolutionary Role and Role in cSCC	Reference
Th1	IL-12, IL-18, IFN-γ, IL-2, IL-28	IL-2, IFN-γ, TNF-α	Intracellular pathogen elimination with activation of microbicidal activity of macrophages. Typically associated with activation of CD8^+^ T cells. *Generally associated with an antitumor role in cSCC.*	[71]
Th2	IL-2, IL-4, IL-33	IL-4, IL-5, IL-6, IL-13, IL-25	Extracellular pathogen and parasitic elimination. Typically, with activation of B cells and humoral responses. *Generally associated with cSCC development.*	
Th9	IL-4, TGF-β	IL-9, IL-10	Extracellular pathogen and parasitic elimination (associates with Th2 responses and may represent a subtype). Associated with allergic inflammation and inflammation in skin. *May play antitumor role in the skin, from mouse models of melanoma (role in cSCC specifically unknown).*	[72]
Th17	IL-1, IL-6, IL-21, IL-23, TGF-β	IL-17, IL-21, IL-22, CCL20, TNF-α	Extracellular pathogen and fungal elimination by enhancing neutrophil responses. *Variably found to be pro- and anti-neoplastic in different models and settings (role in cSCC specifically unknown).*	[73]
Th22	IL-23, IL-6	IL-22, TNF-α	May drive epithelial innate immune responses. Tissue repair post-injury—induces keratinocyte proliferation. *May drive tumor proliferation in cSCC.*	[74]
Tfh	IL-6, IL-21, TGF-α	IL-17, IL-21	Regulation of antigen-specific B cell responses and antibody production through germinal center B-T cell interaction. *Role in cSCC unknown (though presumed to act indirectly through B cell functions).*	[75]
Treg	TGF-β, IL-2	IL-10, TGF-β	Regulatory: induction of tolerance and T cell anergy. Suppression of effector responses. *Suppresses tumor immune responses in cSCC.*	[71]

**Table 2 ijms-20-02009-t002:** Summary of the main classes of immunosuppressant drugs, their main effects on the immune system and additional mechanisms at play in cSCC pathogenesis. 6-MP, 6-mercaptopurine; Aza, azathioprine; IL, interleukin; IMDPH, inosine monophosphate dehydrogenase; mTOR, mammalian target of rapamycin; MMF, mycophenolate mofetil; TGF-β, transforming growth factor-β; UV, ultraviolet; and VEGF, vascular endothelial growth factor.

Class (Drugs)	Effects on Immune System	Additional Mechanisms in cSCC
Calcineurin inhibitors (ciclosporin, tacrolimus)	Bind to intracellular proteins, called immunophilins, to block the effect of calcineurin, which results in reduced production of IL-2 and reduced proliferation of T cells [154]. IL-2 is needed for T cell activation and expansion [155]. Indirectly inhibit monocyte function by suppressing production of γ-IFN, macrophage inhibitory factor, and macrophage chemotactic factor [155]. As a result, IL-l production by monocytes is inhibited. IL-l is a cofactor for activation of T-helper lymphocytes.	Inhibit DNA repair mechanisms, acting synergistically with UV in DNA damage [116,156]. Stimulate tumor growth via VEGF-mediated angiogenesis [157]. Increased invasiveness of cells via TGF-β production [158].
Purine analogues (azathioprine, mycophenolate mofetil)	Aza: prodrug which is converted to 6-MP and metabolized to cytotoxic thioguanine nucleotides, which are responsible for immunosuppression, inhibiting DNA synthesis and inducing apoptosis. Inhibits proliferation of all leucocytes [159,160]. MMF: prodrug of mycophenolic acid, which inhibits purine synthesis by inhibiting IMDPH. Preferentially suppresses T and B lymphocytes [161].	Aza: direct mutagenic effects on DNA and acts synergistically as a chromophore with UV-A to increase sensitivity of cells to DNA damage [152,153].
mTOR inhibitors (sirolimus, everolimus)	Block signaling of the mTOR serine/threonine protein kinase, which suppresses cytokine-driven T-lymphocyte proliferation and activation [154]. Also impairs dendritic cells (DCs), B cells, NK cells, neutrophils [162].	Reduce risk of cSCC: Inhibit angiogenesis by suppressing VEGF [157]. Promote memory T cell function and promote autophagy-mediated DNA repair [116].

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
