# Peer review of "The Role of the Immune System in Cutaneous Squamous Cell Carcinoma"

_ijms, 2019, doi:10.3390/ijms20082009_

Round 1
Reviewer 1 Report
This is an interesting review about immune regulation in cutaneous squamous cell carcinoma. The description in which intensive study of the TME across cancer s including the cSCC has begun uncoupling the complex, dynamic interaction between tumors and the multitude of host innate and adaptive immune cells may be revised to include the discussion how TME regulates the interactions between tumor and immune cells.
Author Response
Reviewer comment 1: This is an interesting review about immune regulation in cutaneous squamous cell carcinoma. The description in which intensive study of the TME across cancer s including the cSCC has begun uncoupling the complex, dynamic interaction between tumors and the multitude of host innate and adaptive immune cells may be revised to include the discussion how TME regulates the interactions between tumor and immune cells.
Response 1: We thank Reviewer 1 for their comments on our review. We have expanded our discussion on how the TME regulates tumour cell -immune cell interactions in the section: "4.3 The Effects of Immunosuppression on The Tumour Microenvironment" and have included references to a few papers that have reviewed this area.
Reviewer 2 Report
Thomson et al. wrote a comprehensive review about the role of the immune system in cutaneous squamous cell carcinoma (SCC). They clearly describe the function of different immune cell populations during SCC development and progression and intensively discuss the role of immunosuppression on SCC development. The manuscript is very clearly written. I have only minor comments.
page 2 lane 45: besides LC, up to 4 dermal DC populations can be identified, which originate from DC-restricted progenitors; in addition monocyte-derived DC an macrophages
page 4 lane 9 and 10: maybe switch IL-22 and iL-33 since ILC2 produce IL-33 and ILC3 produce IL-22
page 4 lane 26 DETC are only found in mice, but there are sparce ggdT cells in human epidermis as well
page 5 lane 13. Explain the abbreviation OTR
Author Response
Point 1: page 2 lane 45: besides LC, up to 4 dermal DC populations can be identified, which originate from DC-restricted progenitors; in addition monocyte-derived DC an macrophages
Response 1: we have clarified that up to 4 dermal DC populations have been identified and added an additional reference.
Point 2: page 4 lane 9 and 10: maybe switch IL-22 and iL-33 since ILC2 produce IL-33 and ILC3 produce IL-22
Response 2: we have amended as suggested by the reviewer.
Point 3: page 4 lane 26 DETC are only found in mice, but there are sparce ggdT cells in human epidermis as well
Response 3: we have amended this section and made a clear distinction between murine and human gamma-delta T cell populations.
Point 4: page 5 lane 13. Explain the abbreviation OTR
Response 4: amended as per reviewer suggestion, thank you.